# Evaluating the Impact of the COVID-19 Pandemic on New Cancer Diagnoses and Oncology Care in Manitoba

Kathleen M. Decker [1,2,3,*], Pascal Lambert [1,3], Allison Feely [3], Oliver Bucher [3], Julian O. Kim [1,4,5], Pamela Hebbard [6], Maclean Thiessen [7,8], Tunji Fatoye [9,10], Marshall Pitz [1,7,8], Rashmi Koul [1,4,5] and Piotr Czaykowski [2,7,8]

1   CancerCare Manitoba Research Institute, CancerCare Manitoba, 675 McDermot Avenue, Winnipeg, MB R3E 0V9, Canada; plambert@cancercare.mb.ca (P.L.); jkim7@cancercare.mb.ca (J.O.K.); mpitz@cancercare.mb.ca (M.P.); rkoul@cancercare.mb.ca (R.K.)
2   Department of Community Health Sciences, Rady Faculty of Health Sciences, Max Rady College of Medicine, University of Manitoba, 750 Bannatyne Avenue, Winnipeg, MB R3E 0V9, Canada; pczaykowski@cancercare.mb.ca
3   Department of Epidemiology and Cancer Registry, CancerCare Manitoba, 675 McDermot Avenue, Winnipeg, MB R3E 0V9, Canada; afeely@cancercare.mb.ca (A.F.); obucher@cancercare.mb.ca (O.B.)
4   Department of Radiology, Rady Faculty of Health Sciences, Max Rady College of Medicine, University of Manitoba, 820 Sherbrook Street GA216, Winnipeg, MB R3T 2N2, Canada
5   Department of Radiation Oncology, CancerCare Manitoba, 675 McDermot Avenue, Winnipeg, MB R3E 0V9, Canada
6   Department of Surgery, Rady Faculty of Health Sciences, Max Rady College of Medicine, University of Manitoba, 820 Sherbrook Street, Winnipeg, MB R3A 1R9, Canada; phebbard@cancercare.mb.ca
7   Department of Internal Medicine, Rady Faculty of Health Sciences, Max Rady College of Medicine, University of Manitoba, 820 Sherbrook Street, Winnipeg, MB R3E 0V9, Canada; maclean.thiessen@umanitoba.ca
8   Department of Medical Oncology and Hematology, CancerCare Manitoba, 675 McDermot Avenue, Winnipeg, MB R3E 0V9, Canada
9   Department of Primary Care Oncology, CancerCare Manitoba, 675 McDermot Avenue, Winnipeg, MB R3E 0V9, Canada; tfatoye@sogh.mb.ca
10  Department of Family Medicine, Rady Faculty of Health Sciences, Max Rady College of Medicine, University of Manitoba, 750 Bannatyne Avenue, Winnipeg, MB R3E 0V9, Canada
*   Correspondence: kdecker@cancercare.mb.ca; Tel.: +1-204-390-3912

**Abstract:** Individuals with cancer are vulnerable to infection with SARS-CoV-2, the virus causing COVID-19. Physical distancing, the reallocation of health care resources, and the implementation of procedures to reduce the spread of COVID-19 may also have serious consequences for people with cancer. We evaluated the impact of COVID-19 on new cancer diagnoses and oncology care in Manitoba, Canada using an interrupted time series design and data from the Manitoba Cancer Registry and CancerCare Manitoba's (CCMB) electronic medical record. In April 2020, there was a 23% decrease in new cancer diagnoses, a 21% decrease in pathology reports, and a 43% reduction in surgical resections. There was no difference in new cancer diagnoses by August 2020, surgery by July 2020, and pathology reports by September 2020. From April 2020 to June 2021, there was a 13% decrease in radiotherapy (RT) fractions, an 18% decrease in UCC visits, and a 52% decrease in in-person visits. There was no change in intravenous chemotherapy visits per month, first RT visits, or overall patient visits. The impact of COVID-19 on shifts in the stage at diagnosis and survival will be assessed in future analyses.

**Keywords:** cancer; COVID-19; epidemiologic studies; surgery; radiotherapy; chemotherapy; Canada

## 1. Introduction

Individuals diagnosed with cancer have proven particularly vulnerable to infection with SARS-CoV-2 infections, the virus that causes COVID-19 [1–4]. Many receive anti-cancer treatments such as chemotherapy that impair their immune systems making them

more susceptible to viruses [4]. Treatments are often complex, requiring multidisciplinary teams to co-ordinate care and communicate clearly and effectively with patients and their families. Physical distancing, the reallocation of health care resources, and the need to implement new procedures to reduce the spread of COVID-19 may also have serious consequences for people with cancer including delays in cancer screening, diagnosis, and treatment [5–10]. In addition to health system changes, patients may also be afraid to access the health care system or experience new barriers to care [11,12]. Therefore, it is important that we understand how COVID-19 has affected the care of individuals with cancer to inform service delivery both during the remainder of the pandemic and in the future. The objective of this study was to evaluate the impact of the COVID-19 pandemic on new cancer diagnoses and oncology care at a system level in Manitoba, Canada from March 2020 to June 2021.

## 2. Materials and Methods

### 2.1. Setting

The province of Manitoba, located in central Canada, has a population of approximately 1.35 million (as of 2021). Two-thirds of the population lives in the capital city of Winnipeg. CancerCare Manitoba (CCMB) is the provincial cancer agency responsible for providing clinical services to all Manitobans diagnosed with cancer. Services, including prevention, early detection, multidisciplinary cancer treatment, supportive and end-of-life care, are provided through four urban locations as well as seven regional and eight community cancer program sites. Approximately 6000 individuals are diagnosed with cancer and 5000 individuals receive treatment or follow-up care at CCMB each year under non-COVID-19 conditions.

Manitoba reported its first COVID-19 case on 12 March 2020. A state of emergency was declared on 20 March 2020 and the first lockdown was implemented on 1 April 2020. By the end of March 2020, patients entering all CCMB locations were screened for COVID-19 symptoms, travel history, and COVID-19 exposure. Entry and exit points were reduced and there was a shift from in-person to telephone or virtual appointments for follow-up and initial consultation visits. Consideration was given to neoadjuvant strategies designed to temporize patients during lockdowns and permit delays in surgery, including approaches such as neoadjuvant systemic therapy for hormone receptor positive breast cancer. Oral therapies and the use of granulocyte-colony stimulating factor as primary prophylaxis to reduce emergency room visits for febrile neutropenia were promoted where appropriate. Non-urgent diagnostic imaging and bloodwork were reduced. Breast screening services were suspended. Hypo-fractionated and single fraction radiation treatments were recommended where possible, to reduce resource consumption and minimise attendance at CCMB [13]. CCMB attendance was also limited to the patient only; no support persons or visitors were allowed except in exceptional circumstances. All patients and staff were required to wear appropriate personal protective equipment (PPE). Elective surgeries were postponed. Radial cancer surgeries were prioritized within disease sites or delayed at the discretion of treating physicians due to a reduction in operation room availability [6].

COVID-19 case counts remained low throughout the spring of 2020 and the province lifted some health orders on 4 May 2020. By late September 2020, Manitoba entered a second wave; a province-wide critical alert was announced on 30 October 2020 and a second lock-down was implemented. Covid-19 cases peaked at 306 per million in November 2020. Health orders were relaxed in February and March 2021. However, new restrictions were announced on 19 April 2021 in response to increasing cases (340.7 cases per million in May 2021) and the start of a third wave of the pandemic. Restrictions were eased in July 2021 as the third wave subsided.

### 2.2. Study Design and Population

We used an interrupted time series (ITS) study design, often used in the evaluation of natural experiments occurring in real-world settings [14], to examine aggregate counts over

time at a system level before COVID-19 (the "pre-intervention" time period: 1 January 2015 until 15 March 2020) and after the start of interventions implemented to address COVID-19 (the "post-intervention" time period: 16 March 2020 until 30 June 2021) on a monthly basis. The study was approved by the University of Manitoba's Health Research Ethics Board (HS23979; H2020:264) and CCMB's Research and Resource Impact Committee (2020-14).

### 2.3. Data Sources

The Manitoba Cancer Registry (MCR) was used to identify new cancer diagnoses and cancer surgical resections. The MCR is a population-based registry that is legally mandated to collect and maintain accurate, comprehensive information about cancer cases in Manitoba and has consistently shown to be of very high quality [13]. MCR data were available from 1 January 2015 to 30 September 2020 for cancer diagnoses and surgery and 31 December 2020 for pathology reports. CancerCare Manitoba's electronic medical record (ARIA MO and RO) was used to determine intravenous (IV) chemotherapy, radiotherapy (RT), Urgent Cancer Care Clinic (UCC) visits, in-person visits, and all visits. ARIA data were available from 1 January 2015 to 30 June 2021.

### 2.4. Outcomes

New cancer diagnoses were defined as the number of new invasive cancer diagnoses (excluding non-melanoma skin cancer and non-Manitoba residents) each month. Pathology reports were defined as the number of reportable pathology reports electronically submitted to the MCR each month. As cancer is a reportable disease, all tissue, blood, bone, and fine needle aspirations in which cancer is suspected must be submitted to the MCR. Surgery was defined as the number of cancer resections performed. IV chemotherapy was defined as the number of IV chemotherapy visits each month (i.e., the number of times patients came to CCMB to receive IV chemotherapy). Radiotherapy treatment was defined as the number of first RT treatment visits each month. RT fractions were defined as the number of days an individual had RT (i.e., the number of fractions) each month. In-person visits were defined as the number of in-person patient visits to a CCMB clinic health care facility and telehealth visits (as individuals are required to visit a health care facility for a telehealth appointment) each month. All visits were defined as the number of in-person, telehealth, telephone, and virtual visits (i.e., telephonic or video visits via Zoom or Microsoft Teams) each month. Urgent Cancer Care clinic (UCC) visits were defined as the number of in-person UCC clinic visits each month. The UCC is located at CCMB's main site and provides care for individuals diagnosed with cancer experiencing acute complications of cancer or its treatment, thereby reducing the need for patients to access hospital emergency department services [15]. If a patient had more than one instance of an individual outcome on the same date and time with the same clinician, the outcome was only counted once.

### 2.5. Statistical Analysis

Generalized linear models (Poisson, quasi-Poisson, negative binomial, gamma, inverse gaussian) were considered for the analyses. Model fit was evaluated by plotting the predicted mean and variance of each model, as well as the observed mean and variance in the data [16]. In addition, scaled quantile residual plots were used to evaluate the overall uniformity of residuals and dispersion [17]. A linear model was also considered, and model fit was evaluated using scaled residual plots. Each model included a binary intervention term that was equal to 0 during the pre-COVID-19 period and 1 during the COVID-19 period, a time term defined as the number of months since the start of the study period, and a month term that accounted for seasonality. Non-linear time and seasonality effects were accounted for using splines. Model building was performed by comparing the adjusted R-squared between subsequent models.

Predicted values to describe the observed data were produced for each of the outcomes using regression models. Counterfactual predictions were also produced for the COVID-19 period by assuming COVID-19 did not occur (hereafter referred to as expected values).

Plots were produced with the observed, predicted, and expected values. If plotted expected estimates were unrealistic (e.g., greatly above what would be expected from the baseline trend), the model was simplified until estimates were consistent with the baseline trend (i.e., the number of degrees of freedom for splines were reduced). COVID-19 by time interactions were considered if plotted predicted values in the COVID-19 period did not fit the observed data well when graphically evaluated (e.g., observed values were below predicted values during the early COVID-19 period but higher during the late COVID-19 period). Because COVID-19 restrictions were implemented incrementally throughout March 2020, this calendar month was excluded from the analyses. Likelihood ratio testing was used to produce *p*-values for the impact of COVID-19 time interactions. Ratios between predicted values and expected estimates, as well as 95% confidence intervals (CI; derived from contrast estimates), were calculated to complement the reported mean differences and 95% CIs. Data analyses were performed in SAS version 9.4 (SAS Institute Inc., Cary, NC, USA) and R version 4.0.4 (R Foundation for Statistical Computing, Vienna, Austria). The following R packages were used: haven, splines, Hmisc, lattice, MASS, ggplot2, car, DHARMa, multcomp, and lmtest.

## 3. Results

Linear models demonstrated better model fit than the generalized linear models and hence, were used to analyze the data. The mean differences and 95% CI from the linear regression results are provided in Table S1. A summary of the ratios between predicted and expected values and 95% CI for each outcome are provided in Table S2. If there was a significant time interaction, the ratios for the April 2020 and the last month of available data are provided. If there was no significant time interaction, an average ratio for the entire time period is shown. Figure 1a shows the observed number of new cancer diagnoses, the predicted number of diagnoses based on the regression model, and the expected number of diagnoses (i.e., the number diagnoses expected in the absence of COVID-19) during the COVID-19 period (grey shaded area). The change in number of cancer diagnoses during the COVID-19 period demonstrated a significant time interaction (*p* value = 0.002). At the start of the COVID-19 pandemic (April 2020), there was a 23% decrease in number of new cancer diagnoses (ratio = 0.77, 95% CI 0.67–0.87). By August 2020, there was no significant difference between the predicted and expected number of new cancer diagnoses (ratio = 0.96, 95% CI 0.87–1.04). Figure 1b shows the number of pathology reports received by the MCR each month, also with a significant COVID-19 by time interaction (*p* value < 0.001). In April 2020, there was a 21% decrease in the number of pathology reports (ratio = 0.79, 95% CI 0.70–0.88). By September 2020, there was no significant difference (ratio = 0.96, 95% CI 0.90–1.02).

A significant time interaction (*p* value < 0.001) was also found for the number of cancer surgical resections (Figure 2a). In April 2020, there was a 43% reduction in the number of surgical resections (ratio = 0.57, 95% CI 0.45–0.70). By July 2020, there was no longer a significant difference between the predicted and expected number of surgical resections (ratio = 0.92, 95% CI 0.83–1.01). Figure 2b shows the number of IV chemotherapy visits per month. There was no significant difference (*p* value = 0.870) between the predicted and expected number of visits (ratio = 1.01, 95% CI 0.91–1.11). There was also no significant difference (*p* value = 0.428) in the number of first RT visits (Figure 2c). The average ratio during the COVID-19 period from April 2020 to June 2021 was 0.96 (95% CI 0.85–1.06). The number of RT fractions was significantly lower (*p* value < 0.001); there was a 13% average decrease in RT fractions during the COVID-19 period (ratio = 0.87, 95% CI 0.81–0.93) (Figure 2d).

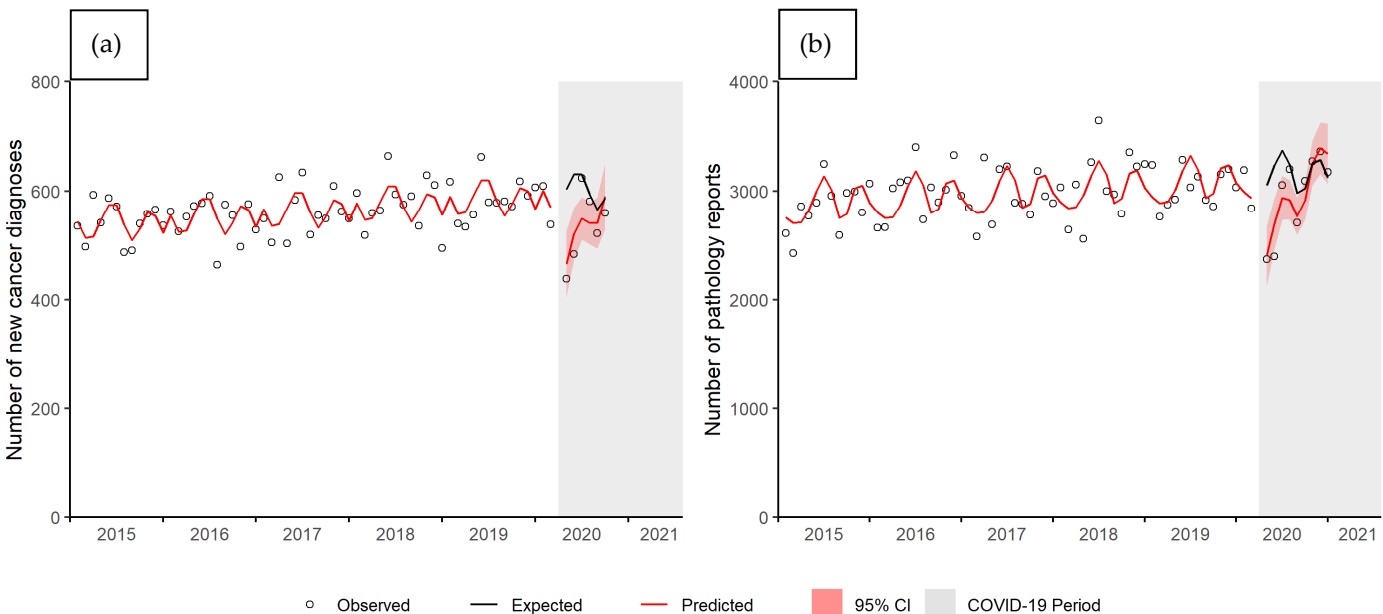

**Figure 1.** Number of (**a**) new cancer diagnoses and (**b**) pathology reports by month, Manitoba. Abbreviations: CI, confidence interval.

Figure 3a shows the number of UCC visits from January 2015 to March 2021. There was a significant (*p* value < 0.001) average 18% decrease in the number of UCC visits (ratio = 0.82, 95% CI 0.75–0.89). In-person visits also significantly (*p* value < 0.001) decreased (Figure 3b). In April 2020, the number of in-person visits decreased dramatically; the average decrease during the COVID-19 period was 52% (ratio = 0.48, 95% CI 0.38–0.58). However, the overall number of patient visits (in-person and telephone) to CCMB did not significantly (*p* value = 0.868) change (ratio = 0.99, 95% CI 0.91–1.08) (Figure 3c).

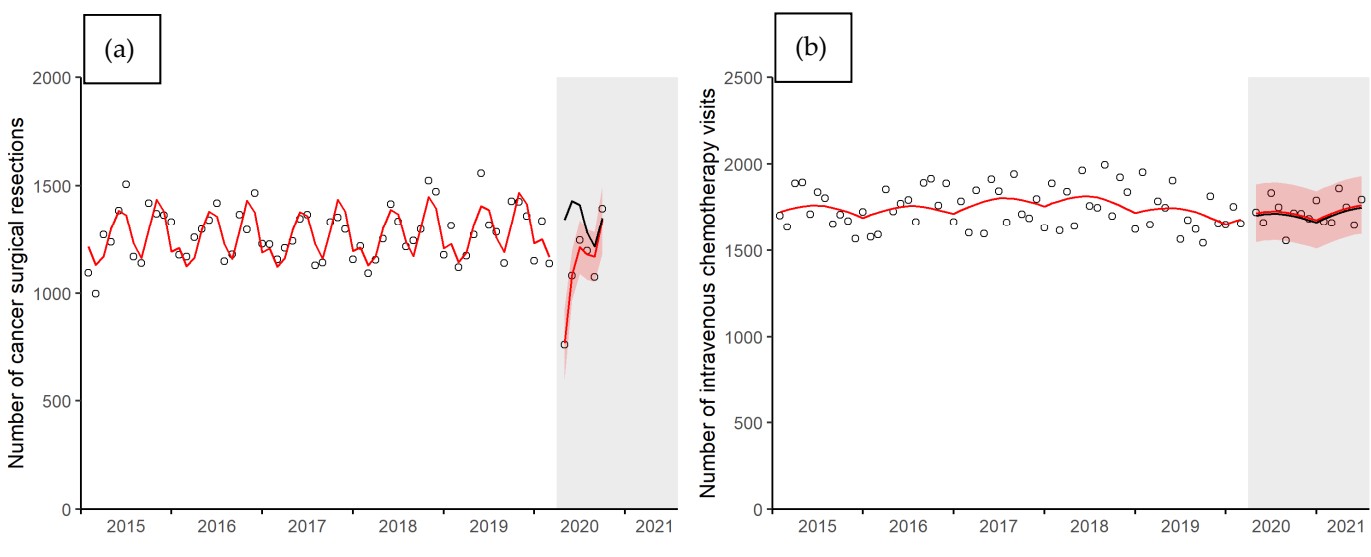

**Figure 2.** *Cont.*

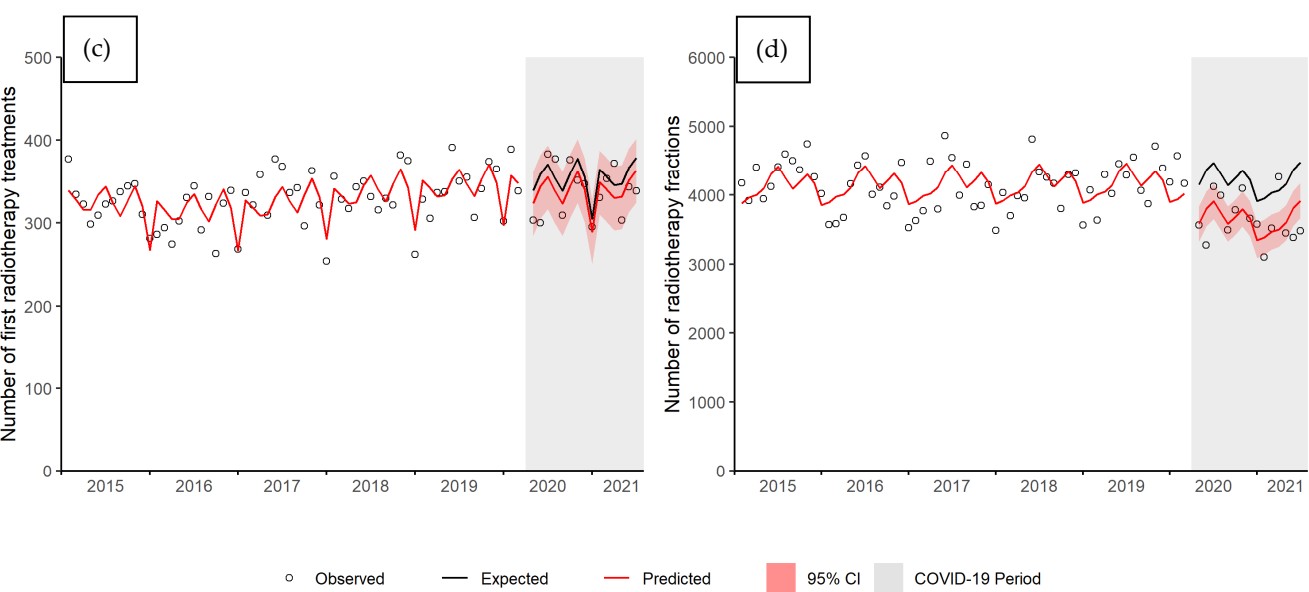

**Figure 2.** Number of (**a**) cancer surgical resections, (**b**) IV chemotherapy visits, (**c**) first radiotherapy (RT) treatments, and (**d**) number of RT fractions by month, Manitoba. Abbreviations: CI, confidence interval.

**Figure 3.** Number of (**a**) Urgent Cancer Care clinic (UCC) visits, (**b**) in-person visits, and (**c**) all visits by month, Manitoba. Abbreviations: CI, confidence interval.

## 4. Discussion

### 4.1. Main Findings

Oncology care in Manitoba rapidly adapted in response to the COVID-19 pandemic with an emphasis on maintaining high standards of care and accessibility while implementing procedures to protect patients and staff. At the start of the pandemic, there was an immediate and substantial decrease in the number of new cancer diagnoses and cancer-related pathology reports. This decrease might be related to challenges associated with the diagnosis of cancer including lower level of primary healthcare utilization, suspended or limited screening programs, and decreases in diagnostic services such computed tomography scans (CTs), biopsies, and surgery. By August 2020, there was no longer a statistically significant difference in the number of cancer diagnoses, and the narrow 95% CI suggest that the numbers were similar to what was expected. In the months following the study period, we may see a rebound in the number of diagnoses to make up for the decreases observed in cancer diagnoses, although this also depends on the impact of the second and third waves of the pandemic on the health care system. One encouraging sign is that the predicted number of pathology reports exceeded the expected number submitted to the MCR during the middle of Manitoba's second wave (December 2020; ratio = 1.07, 95% CI 0.98–1.16) While not significant, this trend may potentially signal a rebound in the number of new cancer diagnoses to follow. However, if a rebound in cancer cases is not observed, it will suggest that there are still individuals who should have been diagnosed but have yet to be, and delays in diagnosis in the near future may be associated with increased mortality for several types of cancer [18]. Because diagnostic and assessment services have yet to reach volumes in excess of pre-pandemic baselines, it is possible that this "lost cohort" of cancer patients may never catch up with the outcome trajectory they would have followed prior to the pandemic. Identifying and defining this cohort will be an on-going challenge moving forward.

There was a 43% decrease in cancer surgical resections in April 2020 which is similar to the 38% fewer cancer surgeries observed in Ontario during the same month and other Canadian provinces [9,19]. Surgery was initially impacted to a greater degree than other forms of cancer treatment because surgical resources, including staff, ventilators, PPE, and hospital rooms for patient recovery, were redirected to address COVID-19 needs. Additionally, some patients may have elected to postpone their surgery and/or begin neo-adjuvant systemic therapy and because there were fewer diagnoses, fewer surgeries may also have been needed. However, the decrease in the number of surgical resections exceeded the decrease in cancer diagnoses by 20% in 2020. It is also important to note that the need to triage cancer surgery was anticipated and planned for at the start of the pandemic in Manitoba [6]. Therefore, some individuals may have experienced different treatment pathways than before COVID-19 which may impact survival. Future analyses will examine the characteristics of these individuals and the impact on survival.

The results of this analysis suggest a gradual recovery in cancer surgery because there was no longer a statistically significant difference in number of cancer surgeries by July 2020. To further catch up, strategies may need to be implemented such as increasing system-wide surgical capacity so that the number of cancer surgeries exceeds the pre-intervention time period (i.e., a ratio above 1). A longer COVID-19 (post-intervention) period is also needed to determine when the number of cancer surgical resections performed returns to expected levels.

There was a non-significant decrease in the number of first RT visits. This decrease likely represents the adoption of recommendations to use radiation judiciously during COVID-19 and avoid radiation in cases where there was minimal or questionable benefits, or delaying radiation until later when temporizing treatments, such as androgen deprivation therapy in prostate cancer, could be provided [13]. The decrease may also be related to fewer diagnoses and decreased availability for surgery which would potentially reduce the demand for neoadjuvant RT. The decrease in the number of RT fractions was in part due to the decrease in first RT visits as well as the implementation of hypo-fractionation regimens.

Before COVID-19, the use of hypo-fractionated RT was highly variable despite data to support its use for common cancer sites, such as prostate, breast, rectum, and lung cancer, as well as for palliative care [13]. The decrease observed in the number of RT fractions correlates with CCMB's wider implementation of hypo-fractionated RT for breast, prostate, lung, brain, and palliative patients, immediately at the start of COVID-19 restriction in line with the most recent guidelines and recommendations [20].

Unlike the trends for surgery or RT, there was no difference at a system level in the number of IV chemotherapy visits per month. This may reflect a relatively small number of new treatment starts compared to the number of patients receiving chemotherapy prior to the start of the pandemic, and the fact that IV supportive treatments were not interrupted. However, it is important to note that this system-level metric may also hide differences in practice at the patient level because some patients may have switched from IV to oral chemotherapy or to a shorter regimen (e.g., eight visits instead of 12) to reduce the need for in-person visits while others, for whom surgery was delayed, may have begun neoadjuvant chemotherapy. An evaluation of the impact of COVID-19 at the patient level on the provision of chemotherapy will be evaluated in a future study.

Similar to reports of reduced primary care clinician and emergency department (ED) visits [21–23], there was a significant and substantial decrease in the number of individuals who visited the UCC. Several studies report that patients have been hesitant to seek medical care out of fear of COVID-19 infection particularly early in the pandemic [11,12]. The decrease in UCC visits may also be related to the decrease in new cancer diagnoses which makes intuitive sense since most UCC visits occur shortly after diagnosis and the start of treatment [15]. This decrease persisted throughout the three waves of the pandemic.

While the number of all visits to CCMB did not change, the number of in-person visits decreased by one-half and were replaced by telephone/video visits. Telephone or video visits were not commonly used prior to the pandemic due to the absence of provincial physician billing codes and privacy concerns, but quickly became integral to the provision of cancer care [6]. These visits have advantages including increased patient convenience and accessibility and lower transportation costs but also potential limitations such as technical issues, inability to complete a full physical examination, and privacy considerations [24]. Additional research is on-going to examine the impact of virtual care on patients' satisfaction and quality of care.

### 4.2. Strengths and Limitations

The results of this study should be interpreted in the context of its strengths and limitations. For this study, we used an ITS study design with a long pre-intervention period which permitted the evaluation of outcomes before the start of the COVID-19 pandemic as well as the inclusion of seasonality and interactions between COVID-19 and time in the analysis. We also had timely access to population-based, system-level cancer data allowing us to quickly disseminate the study findings and help to inform cancer care in Manitoba.

The results must be interpreted in the Manitoba context of COVID-19. Manitoba experienced the easing of restrictions within six weeks of the start of the first wave with very few cases. The second (November 2020 to January 2021) and third waves of the pandemic (April 2021 to June 2021) were significantly more severe. This paper focuses on the measurable impact of COVID-19 on cancer cases and surgical resections during the first wave of the pandemic but includes pathology reports through the second wave and all other indicators throughout all three waves of the pandemic. Therefore, the second and third waves could potentially be associated with larger perturbations in the number of new cancer diagnoses and surgical resections. However, the subsequent waves benefited from advanced health planning, including greater preservation of diagnostic and cancer screening services. The shifting of surgical resources to provide COVID-19-related care was also more pronounced during the subsequent waves and the impact on new cancer diagnoses and cancer care is yet to be measured. Finally, the cumulative impact of recurring shutdowns may or may not have additive effects.

## 5. Conclusions

In Manitoba, the COVID-19 pandemic resulted in a substantial decrease in new cancer diagnoses, pathology reports, and cancer surgical resections immediately after the implementation of COVID-19 restrictions. RT fractions, UCC clinic visits, and in-person visits also decreased. Intravenous chemotherapy visits, first RT visits, and all visits did not change. We will next assess the impact of COVID-19 on patient-level outcomes including potential shifts in the stage at diagnosis and survival in Manitoba.

**Supplementary Materials:** The following are available online at https://www.mdpi.com/article/10.3390/curroncol28040269/s1, Table S1: Linear regression results; Table S2: Ratios and 95% confidence intervals (CI) between predicted and expected values.

**Author Contributions:** Conceptualization, K.M.D., P.L., O.B., J.O.K., M.T., M.P., R.K. and P.C.; data curation, P.L., A.F. and O.B.; formal analysis, K.M.D., P.L., A.F. and O.B.; funding acquisition, K.M.D., P.L., O.B., J.O.K., M.T., M.P., R.K. and P.C.; methodology, K.M.D., P.L., A.F., O.B., J.O.K., M.P. and P.C.; project administration, K.M.D.; supervision, K.M.D.; writing—original draft, K.M.D., P.L., A.F. and O.B.; writing—review and editing, K.M.D., P.L., A.F., O.B., J.O.K., P.H., M.T., T.F., M.P., R.K. and P.C. All authors have read and agreed to the published version of the manuscript.

**Funding:** This research was funded by Research Manitoba and the CancerCare Manitoba Foundation.

**Institutional Review Board Statement:** The study was conducted according to the guidelines of the Declaration of Helsinki, and approved by the University of Manitoba's Health Research Ethics Board, Manitoba Health's Health Information and CancerCare Manitoba's Research and Resource Impact Committee (project code: HS23979 (H2020:264), approval date: 5 May 2020). Because data were de-identified, informed consent was not required.

**Informed Consent Statement:** Patient consent was waived because data were de-identified.

**Data Availability Statement:** The data that support the findings of this study are not publicly available to ensure and maintain the privacy and confidentiality of individuals' health information. Requests for data may be made to the appropriate data stewards (CancerCare Manitoba's Research and Resource Impact Committee).

**Acknowledgments:** We gratefully thank Research Manitoba and the CancerCare Manitoba Foundation for supporting this research.

**Conflicts of Interest:** The authors declare no conflict of interest. The funders had no role in the design of the study; in the collection, analyses, or interpretation of data; in the writing of the manuscript; or in the decision to publish the results.

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
