# Peer review of "Evaluating the Impact of the COVID-19 Pandemic on New Cancer Diagnoses and Oncology Care in Manitoba"

_curroncol, doi:10.3390/curroncol28040269_

Round 1
Reviewer 1 Report
The work of Decker et all is very well structured and implemented. The authors analyze the situation in Manitoba regarding some aspects related to cancer patients during Covid-19.
Some revisions are required in order to improve the quality of their work.
Introduction
-Please add in the aim, at the end of this paragraph, the time considered for this work
Methods
In the paragraph Statistical Analysis for example "Linear models demonstrated better model fit than the generalized linear models and hence, were used to analyze the data." this sentence is a result that comes from the fitting to choose the best model.
Please revised the paragraph and leave there only the methodology. All the passages and the choices made to found the best model or something else please it could be better to report in the results section as a consequence of all analysis planned and implemented.
Results
-Is it possible to add a table with some descriptive statistics on all the selected outcomes? Because only the Figures are not enough to get an idea of ​​these phenomena.
-In Figure 1, is it possible to somehow "improve" the image quality for the last period (2020-2021? It is quite difficult to clearly highlight all the objects and the trend -Please could you also report pvalue as "a significant term ..."
-How did the authors from a clinical-epidemiological point of view manage a significant interaction? Please explain in more detail.
Discussion
-Is there the possibility to add other references related to cancer and covid for other states of North America or in the World to compare results and evidence found and give more generalizability?
Minor
-Please check that the seventh reference is missing something
Reviewer 2 Report
This is a thought-provoking manuscript regarding cancer diagnosis during the first wave of the COVID-19 pandemic. Although the work touches on an important subject the paper is not innovative compared to other previously published studies addressing the same topic.
- I recommend data collection of the second and third wave of pandemic, and comparisons of the different COVID-19 waves would shed light on a subject that has been less studied and would prove meaningful and novel.
- According to the manuscript, IV chemotherapy is the only variable that did not significantly change during the first wave of COVID-19 pandemic. I suggest elaborating on the distribution of the visits between administration of chemotherapy treatments (first treatment versus continuation of treatments), and supportive IV treatments,. Was there more hesitancy of patients to enter the hospital setting and as expected, anticipated lower visitations for non-treatment related admissions?
- Additionally, it would be interesting to present the number of telephone oncology encounters, and has the number changed following the first wave.
Round 2
Reviewer 2 Report
Thank you for the replay and corrections.
Good luck.